# Effect of Adding De-Oiled Kitchen Water on the Bioconversion of Kitchen Waste Treatment Residue by Black Soldier Fly Larvae

**DOI:** 10.3390/ijerph20032577

**Published:** 2023-01-31

**Authors:** Zhuojun He, Cheng Yang, Yan Peng, Taoze Liu, Zhanghong Wang, Chengcai Xiong

**Affiliations:** 1College of Eco-Environmental Engineering, Guizhou Minzu University, Guiyang 550025, China; 2Research Center of Solid Waste Pollution Control and Recycling Engineering, Guizhou Minzu University, Guiyang 550025, China; 3Zunyi Meijing Technology Co., Ltd., Zunyi 563000, China

**Keywords:** black soldier fly larvae, de-oiled kitchen water, sorghum straw, bioconversion, kitchen waste treatment residues

## Abstract

With the continuous development of society, the output of kitchen waste (KW) is fast increasing. De-oiled kitchen water (DKW) and kitchen waste treatment residue (KWTR), two main by-products of the KW treatment industry, are produced accordingly on a large scale. The need to develop an effective technique for the utilization of DKW and KWTR is attracting wide attention. In the present study, black soldier fly larvae (BSFL) were employed as a biological treatment method to treat KWTR with the addition of DKW. The influence of DKW (0–140 mL) on the efficiency of BSFL treatment evaluated by the growth and development of BSFL, the body composition of BSFL, the nutrient content of bioconversion residue (BR), and the bioconversion efficiency of KWTR, was investigated. The results showed that the growth and development of BSFL, the body composition of BSFL, and the conversion rate of KWTR were initially promoted and then inhibited with the addition of DKW. Notably, the amount of DKW added in the T110 group was the most suitable for the growth of BSFL and the accumulation of body composition. Compared with the blank comparison group, the content of crude protein (CP), crude ash (CA), salinity, total phosphorus (TP), and dry matter (DM) of BSFL in the T110 group increased by 3.54%, 6.85%, 0.98%, 0.07% and 2.98%, respectively. However, the addition of DKW could steadily increase the nutrient content of BR, with the highest amount at 140 mL DKW. Following DKW addition, the contents of CP, ether extract (EE), crude fiber (CF), organic matter (OM), total nitrogen (TN), TP, and total potassium (TK) were increased by 4.56%, 3.63%, 10.53%, 5.14%, 0.73%, 0.75%, and 0.52%, respectively, compared with those of the blank comparison group. The study showed that DKW could be used as a nutrient additive in the bioconversion process of KWTR by BSFL, which provided a new method for the resource utilization of DKW.

## 1. Introduction

Kitchen waste (KW), as a sort of residue in the process of processing and eating, accounts for about 30~50% of global solid waste [1,2]. According to the World Food and Agriculture Organization (FAO), about 1.3 billion tons of food are wasted globally every year. A large amount of wasted food produces a huge amount of KW. How to effectively and safely disposed this KW has become a global problem [3,4]. In China, about 75% of KW currently undergoes the pretreatment process of “sorting–crushing–pulping–high-temperature cooking and separation” [5]. After pretreatment, KW is divided into three types of waste: kitchen waste treatment residue (KWTR), de-oiled kitchen water (DKW) and waste oil. Among them, KWTR and DKW contain a high amount of organic matter (OM); the main components of OM include starch, protein, and cellulose [6]. Therefore, KWTR and DKW had great potential for resource utilization. However, in the actual treatment, KWTR and DKW were treated separately. KWTR was used widely for anaerobic gas production [7] and aerobic composting [8], because resources such as biogas and organic fertilizer could be easily obtained. Cudjoe et al. [9] examined the utilization of biogas energy in China in recent years; they found that the contribution of biogas hydrogen production to China’s estimated energy demand was 0.13%, which was equivalent to reducing coal consumption by about 2.10%. However, the compositions of KWTR were different across different regions and time zones, which made the stability of resource products produced by anaerobic gas and aerobic compost not ideal, restricting the development of related industries [10]. For DKW, there were few resource utilization technologies, and the main treatment means was to discharge DKW into sewage treatment plant for decontamination [11]. The purpose of this treatment method was to reduce the pollutant content in DKW. It helped DKW to meet the emission standards. However, protein, ammonia nitrogen and other nutrients in DKW resulted from a lack of resource utilization. Furthermore, in these KW treatment processes, KWTR and DKW were regarded as different wastes and inputted into different treatment units for treatment, which required taking up a lot of land and buying more equipment. Therefore, we attempted to find a simple and efficient method for collaborative resource utilization of KWTR and DKW to reduce the processing cost of KWTR and DKW.

At present, the best potential KWTR resource utilization method was bioconversion by insects. As a method to simulate natural cycle, this method utilized the special mouthparts and digestive system of saprophytic insects to ingested nutrients in KWTR and turned nutrients into energy to maintain insect’s life activities, it could produce high-protein insects; these high-protein insects could replace parts of protein sources and alleviate increasing feed costs and the world food crisis [12]. In fact, in recent years, countries around the world have successively formulated relevant policies on the use of insects as protein sources [13,14]. The promulgation of these policies all indicated that the insect protein market was gradually being taken seriously.

The selection of insects was one of the key factors in bioconversion. At present, a variety of insects have been used in the recycling of waste. For example, Kutty et al. [15] used *Hermetia illucens* L. to degrade KW. Bordiean et al. [16] fed the *Tenebrio molitor* with cheap agricultural and sideline products such as rapeseed meal, wheat bran and sunflower. Zanten et al. [17] used poultry feces and KW as mixed substrates to feed house fly larvae (larvae of *Musca domestica)* to explore the degradation effect of mixed materials by house fly larvae. Among them, the studies of black soldier fly larvae (BSFL) were the most popular, because BSFL had incomparable advantages over other insects, such as natural protection from light, wide recipes, strong resistance to salt and stress [18], as a dominant species, BSFL could effectively inhibit mosquitoes [19] and malodorous gases [20]. In the larval stage, BSFL had weak action ability and huge intake. During the process of digestion, BSFL could convert the animal-derived protein that might exist in the KW into its own specific protein, thus avoiding homology pollution. [21]. In the adult stage, the oral organs of black soldier fly degenerated, and the black soldier fly had a short life cycle and naturally avoided humans [22].

Nowadays, many studies on bioconversion by BSFL have been carried out. These include studies on the type of substrate, the pretreatment of substrate, the conditions bioconversion and the products of bioconversion Singh et al. [23] found that the reduction rate and conversion rate of BSFL to the waste mixed with fruit, vegetables, rice and meat were ideal, which could reach 72% and 25%, respectively. Somroo et al. [24] used the *Lactobacillus buchneri* to ferment bean curd residue, then used BSFL to degrade the fermented bean curd residue, and found that the accumulation of crude protein (CP) and ether extract (EE) of BSFL could be significantly increased by feeding it with fermented bean curd residue. Klüber et al. [25] found that the intestinal microbial community of the BSFL fed with palm kernel meal was mainly *Klebsiella*, *Enterococcus* and *Sphingobacter*. Harnden et al. [26] explored the effect of temperature on bioconversion of mixture of KW by BSFL, and found that 27 °C to 30 °C was the most suitable survival temperature for BSFL. Ratti et al. [27] thought that feeding *Oncorhynchus mykiss* with black soldier fly prepupae instead of some fish meal would not have a negative impact on *Oncorhynchus mykiss*.

However, too little work has been devoted to the effect of the application of DKW to the bioconversion of KWTR by BSFL; therefore, in order to explore the feasibility of application of DKW to the bioconversion of KWTR by BSFL, we planned to choose the mixture of KWTR and sorghum straw powder as substrate, and add quantitative DKW before and during bioconversion to explore the effect of this method on the growth and development of BSFL, body composition of BSFL, nutrient content of bioconversion residue (BR) and bioconversion effect of KWTR. Finally, it provided a new method for the resource utilization of DKW.

## 2. Materials and Methods

### 2.1. Experimental Materials

Eggs of BSFL were obtained from a BSFL breeding company in Zunyi City, Guizhou Province, China.

KWTR was obtained from a kitchen waste treatment plant in Baiyun District, Guiyang City, Guizhou Province, China. The basic physical and chemical properties are shown in Table 1, and the state is deoiled and dehydrated slurry.

DKW was obtained from a kitchen waste treatment plant in Baiyun District, Guiyang City, Guizhou Province, China. The basic physical and chemical properties are shown in Table 2.

Sorghum straw powder was obtained from a sorghum planting base in Renhuai City, Guizhou Province, China. It is dried to a moisture content of less than 10% and then crushed by a high-speed mill.

### 2.2. Experimental Design

The eggs of BSFL were placed in a ventilated hatching box (temperature 28 ± 2 °C) for 36 h and then eggs of BSFL turned into larvae. Larvae were transferred to moist wheat bran (The bran had been fermenting for 3 days, moisture content 60–65%). Larvae were kept for 3 days (The average body length, average body width and average body weight of BSFL reached 4.41 mm, 1.25 mm and 0.002 g, respectively, at the end of the 3 days). In total, 5 g 3-day old BSFL with the same size was screened by sieve. BSFL were placed into a feeding pot (smaller diameter 410 mm × 310 mm × 150 mm, upper diameter of 520 mm × 400 mm × 150 mm) that contained 4000 g KWTR + 250 g sorghum straw powder. The blank comparison group (CK group) (adding 800 mL tap water to the feeding pot before bioconversion, and no water source added during bioconversion), T1 group (adding 800 mL DKW to the feeding pot before bioconversion, and adding no water source during bioconversion), T50 group (adding 800 mL DKW to the feeding pot before bioconversion, and adding 50 mL DKW to the feeding pot every other day during bioconversion), T80 group (adding 800 mL DKW to the feeding pot before bioconversion, and adding 80 mL DKW to the feeding pot every other day during bioconversion), T110 group (adding 800 mL DKW to the feeding pot before bioconversion, and adding 110 mL DKW to the feeding pot every other day during the bioconversion), T140 group (adding 800 mL DKW to the feeding pot before bioconversion, and adding 140 mL DKW to the feeding pot every other day during the bioconversion) were set, respectively. Experiments were replicated three times; each replicate used identical experimental procedures and conditions.

### 2.3. Measurement Indicators

During the bioconversion period, body length, body width, body weight of the BSFL and pH of KWTR (soil acidity meter) were recorded every other day (five BSFLs were randomly selected from each pot and put back after recording).

On the 14th day of bioconversion, BR and BSFL were separated. After drying and crushing, the body composition content of BSFL (CP, EE, crude ash (CA), salinity, total phosphorus (TP), calcium and dry matter (DM)), BR nutrient content (CP, EE, crude fiber (CF), OM, pH, total nitrogen (TN), TP, and total potassium (TK)) along with the effect of bioconversion of KWTR (reduction rate, conversion rate and output of BSFL) were determined.

### 2.4. Analytical Methods

#### 2.4.1. Formulas Involved in the Experiment

The formulas involved in the article are as follows:$$\mathrm{Reduction}\;\mathrm{rate}\;\mathrm{of}\;\mathrm{KWTR}=\frac{W_{1}-W_{2}}{W_{1}}\times 100\%$$
$$\mathrm{Conversion}\;\mathrm{rate}\;\mathrm{of}\;\mathrm{KWTR}=\frac{W_{6}-W_{4}}{W_{1}-W_{2}}\times 100\%$$
$$\mathrm{Average}\;\mathrm{daily}\;\mathrm{weight}\;\mathrm{gain}\;\mathrm{of}\;\mathrm{BSFL}=\frac{W_{5}-W_{3}}{13}$$
$$\mathrm{Output}\;\mathrm{of}\;\mathrm{BSFLwet}\;\mathrm{basis}=W_{5}-W_{3}$$
$$\mathrm{Output}\;\mathrm{of}\;\mathrm{BSFLdry}\;\mathrm{basis}=W_{6}-W_{4}$$

*W*_1_ is the weight (dry basis) of KWTR before bioconversion, *W*_2_ is the weight (dry basis) of KWTR after bioconversion, *W*_3_ is the weight (wet basis) of BSFL before bioconversion, *W*_4_ is the weight (dry basis) of BSFL before bioconversion, *W*_5_ is the weight (wet basis) of BSFL after bioconversion, and *W*_6_ is the weight (dry basis) of BSFL after bioconversion.

#### 2.4.2. Index Measurement Method

The initial characterization of BSFL and BR was performed on a dry weight basis. CP was calculated as analyzed TN × 6.25 [28]. TN and TP. The mixed sample of dried and crushed BSFL, 5 mL H_2_SO_4_ and 2 g CuSO_4_ was placed into the cooking oven and cooked for 150 min; the automatic analyzer (Cleverchem 380, DeChem-Tech. GmbH, Hamburg, Germany) was then used for determination. EE: The sample was degreased by the petroleum ether for four hours using the Soxhlet extraction device; the weight difference before and after degreasing was the weight of the EE [29]. CF: After degreasing, a 2 g sample was mixed with 200 mL H_2_SO_4_ at a concentration of 12.5 mg/mL and heated for 30 min, washed with distilled water until neutral, then mixed with 200 mL NaOH with concentration of 12.5 mg/mL and heated for 30 min, washed until neutral and dried for determination of weight. Then, it was burned in a 600 °C high temperature furnace for 30 min, the weight of ash was obtained. Finally, the CF content could be obtained by difference of weight [30]. Crude ash: The sample of BSFL was broken and dried and then burned in a high temperature furnace at 550 °C for 3 h, so that the weight of the samples did not change before being taken out and weighed [31]. OM: The sample of BSFL was broken and dried, weighed, and then burned in a high temperature furnace. Recording weight of the samples after burning and taking the weight difference [32]. TK and Calcium: A 2 g sample was mixed with 10 mL mixture of hydrochloric acid (HCl) and water (HCl:water =1:3), diluted the mixed sample to 100 mL, then put 50 mL filtrate into a flame photometer for determination [33]. Moisture: This was determined by the gravimetric difference of the final weight before and after drying [34]. Salinity: A 2 g sample was burned in 550 °C, 0.5 g ash of sample, 250 mL distilled water and a few drops of K_2_CrO_4_ indicator were mixed, then titrated by AgNO_3_ [35].

### 2.5. Data Processing

Excel 2020 was used for the process of treatment of experimental data; all results are presented as mean ± SD. Origin 2018 was used for charting. SPSS 17.0 (IBM, Armonk, NY, USA) was used for significant difference analysis, and comparisons among treatments means were performed using Tukey’s test with effects declared significant at *p* < 0.05.

## 3. Results and Discussion

### 3.1. Effect of Adding DKW on Growth, Development and Body Composition of BSFL 

The growth and development of BSFL is summarized in Figure 1, Figure 2 and Figure 3. According to the changing trends in body length, body width and body weight of BSFL, it could be found that during 3 to 7 days of bioconversion, the BSFL of T1 group was lower than that of most experimental groups in terms of body length, body width, and body weight. This might be caused by the unreasonable manner of adding DKW to the T1 group. First, at the beginning of bioconversion, BSFL was transferred from the nursery material into the substrate of the T1 group. At this time, the liquid composition of the nursery material was mainly tap water, whereas the liquid composition of T1, T50, T80, T110 and T140 was mainly DKW. Different liquid components inhibit the growth and development of BSFL. This might be related to the intestinal microbial community of BSFL. Some studies showed that when the feeding environment changed, the intestinal microbial community of BSFL also changed; the change in the intestinal microbial community of BSFL was to help BSFL adapt to the new feeding environment [36,37]. On the 5th day of bioconversion, the body weight of BSFL in the T50, T80, T110 and T140 groups was significantly heavier than that in T1 group, which might be caused by two reasons. On the one hand, with the growth and development of BSFL, the intestinal microbial community of BSFL and DKW formed a new niche; at this time, the inhibition of DKW on BSFL disappeared. On the other hand, the reducing sugar, amino acid, fatty acid and other small soluble substances in DKW [38] began to be absorbed and utilized by BSFL. Therefore, the growth and development of BSFL in T50, T80, T110 and T140 groups were accelerated, in the middle and late stage of bioconversion, the growth level of these BSFL with continuous addition of DKW were the same as that of CK group. However, the growth and development of BSFL in T1 group had been at the lowest level in the whole process of bioconversion. Because there was no water source added during whole process of bioconversion, and the DKW in the early substrate stimulated the intestinal microorganisms of BSFL and inhibited the growth and development of BSFL. From the body length, body width and body weight of BSFL on the 13th day of bioconversion, the continuous addition of DKW (T50, T80, T110 and T140 groups) would not have a significant impact on the growth and development of BSFL. Table 3 presents the relevant indicators regarding the yield and average daily weight gain (ADWG) of BSFL. It was worth noting that during the whole bioconversion process, on the one hand, BSFL showed inhibited growth and development by DKW at the initial stage of transformation (the intestinal microbial community of BSFL changes due to different water sources). On the other hand, BSFL promoted growth and development by DKW at the middle and late stages of bioconversion (nutrients in DKW were absorbed and utilized by BSFL). In the growth and development of BSFL, the dominant influences could be judged by ADWG. The T80 and T110 groups were significantly larger than the CK group, indicating that under these two kinds of methods (T80 and T110 groups), DKW finally promoted the body weight of BSFL. In the output of BSFL (wet basis) and the output of BSFL (dry basis), the T80 and T110 groups were significantly heavier than the CK group, which reflected that DKW promoted BSFL not only in solid components (such as CP, EE, etc.), but also in liquid components (such as water of in body, etc.).

The body compositions of the BSFL are summarized in Table 4. In this work, the CP content of BSFL found in the T1 group was significantly smaller than in the CK group. The T80, T110, and T140 groups were significantly larger than the CK group, but there was no significant difference among the T80, T110, and T140 groups. Compared with the CK group, the T110 group increased by 3.54%. A study by Makkar found [39] that the CP content of BSFL was usually between 39.9% and 43.1%; the CP content of CK group in our study conformed to this range, but the T80, T110, T140 groups were larger than this range, which indicated that the adding DKW could significantly increase the CP content of BSFL. Concerning the EE content, the T50, T80 and T140 groups were significantly smaller than the CK group, but the T1, T110 and CK groups showed no significant difference, which might be related to the CF content of sorghum straw in the feeding substrate. Wu et al. [40] found that the intake of CF could promote the decomposition of carbohydrates, fatty carbon and polysaccharides in gut of BSFL. Zheng et al. [41] found that the intake and degradation of CF by BSFL increased with the increase in microbial content on the substrate. We used DKW to soak sorghum straw, which could greatly enhance microbial richness of substrate and promote intake of CF. Therefore, BSFL at T50, T80, and T140 groups had lower than the CK group in EE content. The BSFL of all groups of adding DKW were significantly smaller than CK group in terms of CA content. The T110 group was the smallest; compared with the CK group, the T110 group decreased by 6.85%. As an oxide produced by high temperature burning, CA was considered by mainstream studies [42] as an index to indirectly judge the OM content of the tested samples. Therefore, low CA content indicated that the BSFL of groups of adding DKW had more OM content. For the salinity content, all groups of adding DKW were significantly larger than the CK group, and the salinity content increased with the increase in added DKW. The high salinity content of the feed was not conducive to the development of the feed processing industry, but this is no great cause for concern. On the one hand, the salinity content of KW of different areas varied, but the overall amount was within a fixed range, about 4% to 6% [43]. On the other hand, because BSFL had an ion balance regulation mechanism, the salinity content absorbed by BSFL was limited. The TP of the T110 group was significantly larger than that of the CK group. Compared with the CK group, the amount in the T110 group was increased by 0.07%; other groups comparing the DKW and CK groups showed no significant difference. By comparing the calcium content of different groups, we determines that all groups showed no significant difference; therefore, it could be speculated that the addition of DKW had no effect on the accumulation of calcium in BSFL. In the determination of DM content, the T50, T80 and T110 groups were significantly heavier than the CK group, and the T1, T140 and CK groups showed no significant difference. This indicated that the continuous addition of DKW (such as T50, T80 and T100 groups) could promote the DM accumulation of BSFL, but the excessive addition of DKW (such as T140 group) would inhibit the DM accumulation of BSFL.

### 3.2. The Effect of Adding DKW on the Nutrient Content of BR

Figure 4 and Table 5 showed the pH changes in KWTR during the bioconversion and the nutrient content data of BR after the end of the conversion. They reflected the effect of DKW on KWTR. It could be seen from Table 5 that with the addition of DKW, the CP content of BR in the T50 and T140 groups was significantly higher than that in the CK group, but there was no significant difference between BR in the T80, T110, and CK groups. However, the T110 group in CP content of BSFL was significantly higher than in the other groups, which indicated that CP in BR was transferred from BR to BSFL. All groups of DKW (T1, T50, T80, T110, and T140 groups) showed no significant difference in EE content, and all of them were significantly larger than the CK group. This might be because the EE content of BR mainly came from the oil contained in DKW and KWTR themselves. However, the oil content of DKW was low after deoiling treatment, generally only 1~5% of the original wastewater [44]. At this time, the EE content of the whole substrate was mainly composed of KWTR, which made no significant difference in EE content of experimental groups adding DKW. The amount of CF in all of the DKW groups showed no significant difference and was significantly smaller than in the CK group. This might be because most of the CF in BR came from sorghum straw powder (the CF content in sorghum straw was about 50~70% [45]), and a small part came from rice and vegetables of KWTR (the CF content in rice and vegetables was about 0.1~3% [46]), CF was not easily degraded by environmental microorganisms under normal conditions; however, long-term immersion in DKW could greatly improve the abundance of environmental microorganisms that produce cellulase [47], so as to improve CF degradation efficiency. In our study, the range of OM content of six experimental groups was between 69.40% and 74.54%. This was similar to the OM content of BR produced by Borkent et al. after the bioconversion of KW by BSFL [48]. The OM content of T80, T110 and T140 groups were significantly larger than CK group, T140 group had the largest OM content of BR, which was increased by 5.14% compared with CK group. Usually, the OM content decreases with the bioconversion, which was mainly the result of the utilization of carbon by environmental microorganisms [49]. Due to the additional carbon source brought by DKW and sorghum straw, however, some environmental microorganisms degrade DKW and sorghum straw, so that OM in KWTR could be retained; therefore, the groups of DKW were larger than CK group in OM content. The pH levels were observed in our study; all experimental groups showed no significant difference and were between 6.70 and 6.95 at the end of the bioconversion, which was quite different from the study result of Awasthi et al. [50]. In their study, the pH of BR was 4.27 at the end of the bioconversion, which might be caused by the difference of oil content. The KWTR we obtained was degreased (oil content was 0.57%), whereas the KW of Awasthi et al. was not degreased. The oil in KW would hinder the substrate and the oxygen exchange of air [51], and caused the pH of BR to decrease. Moreover, the pH of BR was neutral and weakly acidic in our study, which might be due to incomplete degradation of organic acids produced by BSFL during bioconversion [52]. The TN content of six experimental groups was between 3.8% and 4.7%. The nitrogen content of the T50 and T140 groups was significantly larger than the CK group, which showed that BSFL’s absorption of nitrogen sources in KWTR and DKW of the T50 and T140 groups was lower than that of other experimental groups, meaning more nitrogen sources from KWTR and DKW remained in BR. Concerning TP content, the T80, T110, and T140 groups were significantly larger than the CK group, which was increased by 0.34%, 0.56%, and 0.75%, respectively, compared with the CK group. It was a remarkable fact that comparing the TP content of BR with the TP content of BSFL, we found that all groups of adding DKW were significantly larger than CK group in BR; however, the groups with added DKW had no significant difference with CK group in BSFL, except for the T110 group. This indicated that the total amount of phosphorus in DKW was mainly retained in BR, and very little TP would be utilized and absorbed by BSFL. The increase in TP in BR was helpful to improve the quality of organic fertilizer. The TK increased with the increase in the DKW, T50, T80, T110, and T140 groups, which increased by 0.11%, 0.17%, 0.31%, and 0.52%, respectively, compared with the CK group. Organic fertilizers rich in potassium could better promote the growth of crops [53]; thus, the addition of DKW was conducive to improving the quality of organic fertilizers prepared by BR.

Through the above analysis, we found that the content of CP, OM, TN, TP, and TK in BR were positively correlated with the amount of added DKW in general. This meant that some part of DKW was not degraded by BSFL; under the action of environmental microorganisms, the DKW could be retained in BR in the form of a solid, which ultimately increased the nutrient content of BR [54]; therefore, the nutrient content of BR after absorbing DKW was higher than that of ordinary BR.

### 3.3. The Effect of Adding DKW on the Bioconversion Effect of KWTR

Figure 5 showed that reduction rates of the T1 and T140 groups were significantly smaller than those of the CK group. The T50, T80, T110, and CK groups showed no significant differences. The T110 group reached 63.73%; this reduction rate was lower than the 71.94% reduction rate of the mixed KW from the findings of Singh et al. [23]. This result might be related to the amount of added DKW. As a high-concentration organic wastewater with complex components, DKW was doped with a large amount of the suspended solids (SS) [55]. Under the combined effect of the increase in DKW and the evaporation of water within the KWTR [56], SS in DKW remained in KWTR as a solid, increasing the weight of KWTR; the reduction rate essentially reflects the weight change of KWTR, when the weight of the residual SS was larger than the intake of KWTR by BSFL, and the reduction rate was decreased. The T80 and T110 groups of the conversion rate were significantly larger than the other experimental groups. The T110 group was largest, reaching 29.79%, and increased by 5.04% compared with the 24.75% conversion rate of mixed KW in the study by Singh et al. The T110 group was much higher than the conversion rate of the study by Singh et al. This might be related to the rich nutrients in DKW. The addition of DKW increased the nutrient intake of BSFL, so that BSFL could obtain a larger level of DM accumulation, and thereby improving the conversion rate of KWTR. For the conversion rate, the added DKW had a significant promoting effect. The suitable range of addition was shown in the T80 and T110 groups; among them, the T110 group had the best effect. 

From the above data, we found that an appropriate addition of DKW could increase the conversion rate of KWTR without significantly reducing the reduction rate of KWTR.

## 4. Conclusions

Adding DKW in the process of decomposing KWTR by BSFL could not only reduce the cost of treatment, but also improve BSFL’s various indicators; thus, it was feasible to add DKW during the bioconversion of KWTR from BSFL. Under the conditions of our experiment, the addition of the appropriate DKW could promote the growth and development of BSFL, the body composition of BSFL, the nutritional composition of BR, and the bioconversion effect of KWTR. The addition of the T110 group made the growth and development of BSFL, the body composition of BSFL, and the bioconversion effect of KWTR rise to the highest level. The body length, body width, body weight, average daily weight gain, output (dry basis), CP, EE, TP, CA, and DM of BSFL of T110 increased by 0.53 mm, 0.05 mm, 0.03 g, 7.47 g/d, 53.17 g, 3.54%, 0.74%, 0.07%, 0.16%, and 0.58%, respectively, compared with the CK group. However, if adding DKW was too high, it would have an inhibitory effect for BSFL. For BR, this was different. The nutrient content of BR increased with the increase in added DKW; among them, the nutrient content of T140 had the highest level, the CP, EE, OM, TN, TP and TK of BR of T140 group increased by 4.56%, 3.63%, 5.14%, 0.73%, 0.75%, 0.52%, respectively, compared with the CK group. In addition, the addition of DKW had a certain adverse effect on the reduction rate of KWTR, compared with the CK group; the production rate of the T110 group increased by 6.89%, but the T110 group and the CK group showed no significant difference. To utilize DKW and produce BSFL as much as possible, the best addition of DKW was from the T110 group.

In the future, we will investigate the optimal additional amount of DKW at different temperatures, the characteristic microorganisms in DKW, the feasibility of adding DKW on a larger scale, and the bioconversion of KWTR by BSFL.

## Figures and Tables

**Figure 1 ijerph-20-02577-f001:**
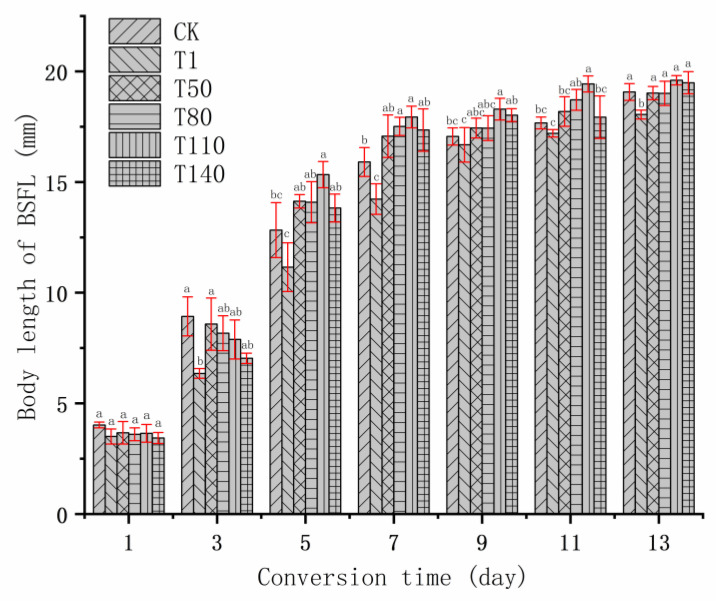
Trend of body length of BSFL. Note: The different lowercase letters indicate significant differences among treatments (*p* < 0.05).

**Figure 2 ijerph-20-02577-f002:**
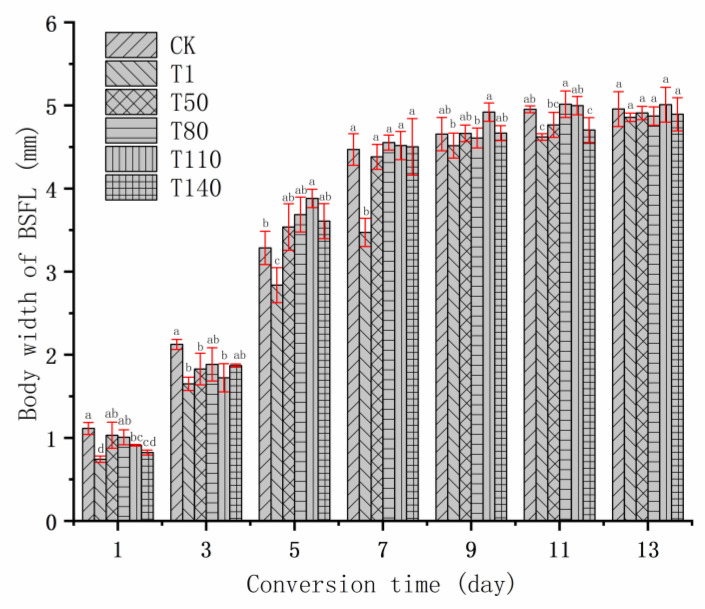
Trend of body width of BSFL. Note: The different lowercase letters indicate significant differences among treatments (*p* < 0.05).

**Figure 3 ijerph-20-02577-f003:**
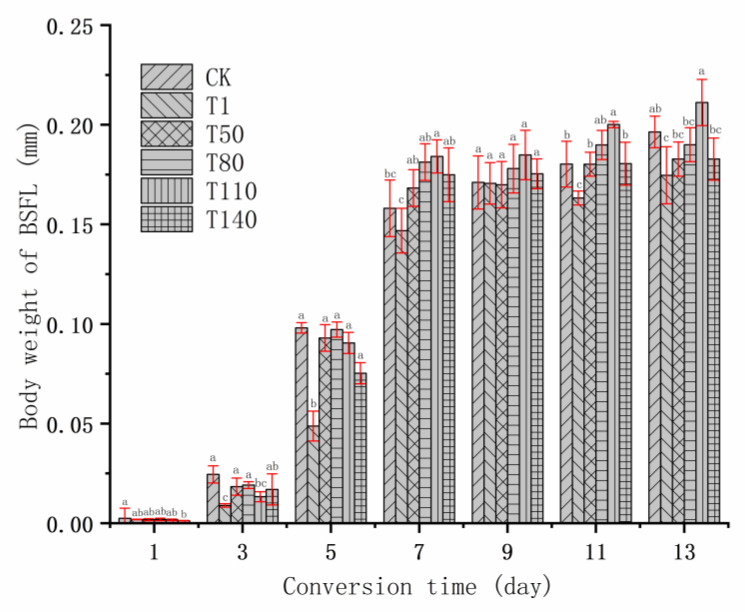
Trend of body weight of BSFL. Note: The different lowercase letters indicate significant differences among treatments (*p* < 0.05).

**Figure 4 ijerph-20-02577-f004:**
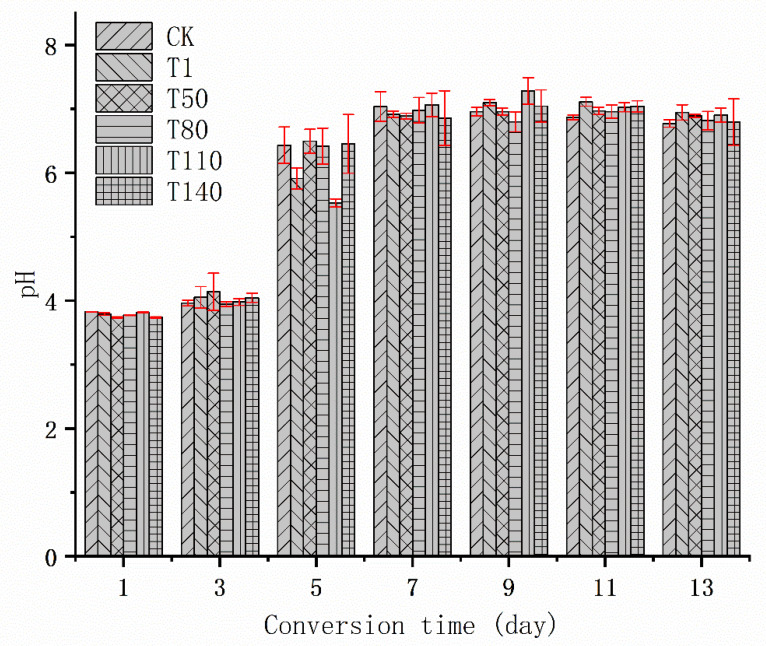
Effect of adding DKW on the pH variation trend of KWTR.

**Figure 5 ijerph-20-02577-f005:**
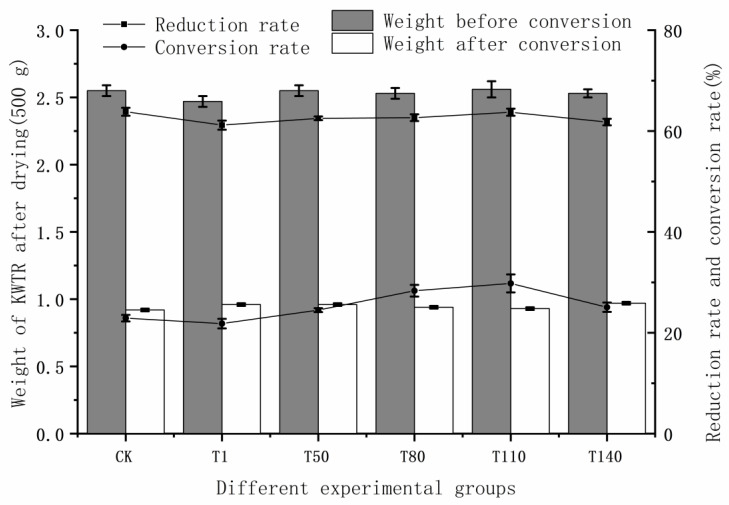
Effect of the amount of DKW on the reduction rate and bioconversion rate of KWTR.

**Table 1 ijerph-20-02577-t001:** Basic physical and chemical properties of KWTR.

Items	Content
Moisture	74.29%
pH	3.92
Crude protein	20.83%
Ether extract	18.30%
Salinity	1.54%
Oil	0.57%

**Table 2 ijerph-20-02577-t002:** Basic physical and chemical properties of DKW.

Items	Content
pH	3.79
NH3-N	42.76 mg/L
Total suspended solids	8331.84 mg/L
Salinity	2.85%
Organic matter	5350.47 mg/L
Oil	0.24%

**Table 3 ijerph-20-02577-t003:** Effect of the amount of DKW on the growth and development of the BSFL.

Item	CK	T1	T50	T80	T110	T140
ADWG/(g/d)	79.74 ± 1.70 c	74.81 ± 0.62 d	76.14 ± 0.43 d	82.18 ± 0.02 b	87.21 ± 1.38 a	79.23 ± 0.96 c
Output (wet basis)/(g)	1036.63 ± 22.06 b	972.59 ± 8.00 a	989.85 ± 5.54 a	1068.36 ± 0.29 c	1133.74 ± 17.91 d	1030.03 ± 12.50 b
Output (dry basis)/(g)	191.53 ± 1.19 c	169.87 ± 2.85 d	200.50 ± 6.93 c	229.20 ± 7.01 b	248.03 ± 6.51 a	200.90 ± 9.69 c

Note: Different shoulder letters in the same industry indicate significant differences at the level of difference < 0.05.

**Table 4 ijerph-20-02577-t004:** Effect of adding DKW on the body components of the BSFL.

Item/%	CK	T1	T50	T80	T110	T140
CP	41.32 ± 1.78 bc	33.24 ± 1.22 a	39.55 ± 1.50 b	44.26 ± 0.78 cd	44.86 ± 0.49 d	43.98 ± 1.57 cd
EE	32.89 ± 0.77 a	32.38 ± 0.53 a	29.16 ± 0.65 c	30.30 ± 0.83 bc	33.63 ± 0.32 a	30.69 ± 0.24 b
CA	26.84 ± 0.58 a	23.67 ± 0.43 b	23.05 ± 1.29 bc	21.99 ± 0.10 c	19.99 ± 0.25 d	22.55 ± 0.49 bc
SAL	1.74 ± 0.042 d	2.43 ± 0.037 c	2.57 ± 0.023 c	2.57 ± 0.006 bc	2.72 ± 0.033 b	2.94 ± 0.046 a
TP	0.87 ± 0.05 a	0.89 ± 0.02 ab	0.91 ± 0.01 ab	0.94 ± 0.01 ab	0.94 ± 0.01 b	0.88 ± 0.00 a
Ca	4.51 ± 0.16 a	4.61 ± 0.24 a	4.81 ± 0.12 a	4.66 ± 0.02 a	4.67 ± 0.1 a	4.76 ± 0.64 a
DM	18.47 ± 0.49 cd	17.47 ± 0.46 d	20.26 ± 0.22 ab	21.45 ± 0.82 a	21.88 ± 0.50 a	19.50 ± 0.47 bc

Note: Different shoulder letters in the same industry indicate significant differences at the level of difference < 0.05.

**Table 5 ijerph-20-02577-t005:** Effect of adding DKW on the content of nutrients in BR.

Item	CK	T1	T50	T80	T110	T140
CP/%	24.25 ± 0.81 c	25.82 ± 0.91 bc	27.22 ± 0.46 ab	26.09 ± 0.98 bc	25.84 ± 1.18 bc	28.81 ± 1.30 a
EE/%	20.23 ± 1.23 b	23.30 ± 0.37 a	23.03 ± 0.61 a	24.67 ± 0.60 a	23.10 ± 1.09 a	23.86 ± 1.11 a
CF/%	28.50 ± 2.57 b	23.60 ± 0.91 a	22.77 ± 1.50 a	21.75 ± 1.78 a	22.07 ± 2.40 a	23.03 ± 1.73 a
OM/%	69.40 ± 2.05 c	71.27 ± 1.27 c	72.53 ± 0.90 bc	72.98 ± 0.50 ab	73.40 ± 0.61 ab	74.54 ± 2.33 a
pH	6.77 ± 0.06 a	6.95 ± 0.12 a	6.89 ± 0.02 a	6.82 ± 0.15 a	6.91 ± 0.01 a	6.80 ± 0.36 a
TN/%	3.88 ± 0.13 c	4.13 ± 0.15 bc	4.36 ± 0.07 ab	4.17 ± 0.16 bc	4.13 ± 0.19 bc	4.61 ± 0.21 a
TP/%	1.13 ± 0.05 d	1.14 ± 0.04 d	1.22 ± 0.03 d	1.47 ± 0.11 c	1.69 ± 0.06 b	1.88 ± 0.03 a
TK/%	0.20 ± 0.010 d	0.22 ± 0.016 d	0.31 ± 0.004 c	0.37 ± 0.011 c	0.51 ± 0.035 b	0.72 ± 0.056 a

Note: Different shoulder letters in the same industry indicate significant differences at the level of difference < 0.05.

## Data Availability

Some or all data that support the findings of this study are available from the corresponding author upon reasonable request.

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
