# Peer review of "Effect of Adding De-Oiled Kitchen Water on the Bioconversion of Kitchen Waste Treatment Residue by Black Soldier Fly Larvae"

_ijerph, 2023, doi:10.3390/ijerph20032577_

Round 1

Reviewer 1 Report

The present research was focused on the treatment of KWTR with the addition of KDW. It is interesting. However, from design to results, the logic of the whole manuscrip is a bit confusing. Furthermore, the author did not discuss the results in depth, and mostly elaborated on the results. So it did not reach the purpose of "provided a new way for the resource utilization of KDW".

1. The whole manuscript needs refinement. For example, the introduction should be refined with the purpose and content of this study. The origin of KWTR and KDW should be introduced together because they were got together. It is not necessary to repeat the results that are clearly visible from the tables and figures, just state the most important findings.

2. Earthworm is not insect (line 80)

3. Material and Method:

(1) Why you mixed 4000 g KWTR with 250 g sorghum straw powder?

(2) During the 14 day’s experiment, the moisture content should be paid more attention.

4. Results

(1) Only Table 3 presented the growth and development of BSFL

(2) The moisture content of CK and T140 is similar. However, in the experimental design, the author put additional 140ml DW every day for 14 days. Generally speaking, the moisture content is obviously different.

(3) The author considered that the composition of environmental microbial communities varied greatly under different feeding substrate which effected the bioconversion. However, the different moisture content might be an important factor for the activity of microbes.

(4) “the feeding substrate of groups of KDW addition (KWTR + sorghum straw + KDW) were different from that the feeding substrate of 1-day old BSFL,” on line 230-231. I could not understand the meaning.

(5) The author pointed out that the salinity inhibited the activity of BSFL, and they may need 1-3 days to maintain the ion balance. However, the author did not determine the salinity during the process. And different addition of KDW will change the salinity of the feeding substrate. So the explanation should be re-considered.

(6) What is the “the amount of KDW” mean in Table 4?

Author Response

Dear reviewer:

Salutation!

First of all, thank you very much for taking time out of your busy schedule to read and revise our article. Thank you for your valuable comments. You have made a comprehensive correction to the structure, content, research methods and results of our paper. It has played a very important role in improving the quality of our article.

We carefully studied the comments of the reviewers and revised the article according to the comments.

 It must be pointed out that about the grammar of the article, we have tried our best to modify it. However, because of different language usage habits and insufficient English level, there may still be some problems in the revised sentence. we will use the article polishing service provided by the journal to further adjust the article.

  1. The whole manuscript needs refinement. For example, the introduction should be refined with the purpose and content of this study. The origin of KWTR and KDW should be introduced together because they were got together. It is not necessary to repeat the results that are clearly visible from the tables and figures, just state the most important findings.

Response: Introduction and Result and Discussion had been rewritten. The main modifications were as follows: the less relevant part in (introduction) and the description of data in (Result and Discussion) had been deleted, and the analysis of experimental results and phenomena in (Result and Discussion) had been added.

  1. Earthworm is not insect (line 80)

Response: The part involving earthworms had been deleted and the relevant content of the bioconversion method of Tenebrio molitor had been added in Introduction.

  1. Material and Method:

(1) Why you mixed 4000 g KWTR with 250 g sorghum straw powder?

Response: Through the pre-experiment, we found that the moisture content of the substrate after the mixture of 4000g of kitchen waste and 250g of straw powder was optimal. The specific explanation was as follows: On the one hand, it could make the black soldier fly larvae develop normally during the bioconversion. On the other hand, it was easier to separate black soldier fly larvae and the bioconversion residue after bioconversion.

(2) During the 14 day’s experiment, the moisture content should be paid more attention.

Response: We added sorghum straw powder in the process of substrate mixing to control the moisture content of the substrate. From the growth and development of the black soldier fly larvae and the bioconversion effect of kitchen waste, the increase of substrate moisture did not seriously affect the bioconversion process.

  1. Results

(1) Only Table 3 presented the growth and development of BSFL

Response: Due to the negligence of the writing process, the figure 1, figure 2 and figure 3 were incorrectly written into table 1, table 2 and table 3. This error had been corrected

.

(2) The moisture content of CK and T140 is similar. However, in the experimental design, the author put additional 140ml DW every day for 14 days. Generally speaking, the moisture content is obviously different.

Response: We think that this phenomenon could be explained from the following three points: â‘  We added 250g of sorghum straw powder to KWTR before the bioconversion. These sorghum straw powder in the conversion process could make the 140 mL KDW we added every other day be absorbed, so as to control the moisture content of the whole substrate. â‘¡ One of the purposes of adding KDW every other day was to let BSFL have sufficient time to absorb KDW, so as to avoid the difficulty of separating BSFL and KWTR due to high moisture content of substrate. â‘¢ Liquid part of KDW volatilized due to the biological activity of BSFL and temperature of environment. Therefore, at the end of bioconversion, the moisture content of bioconversion residue in CK group was similar to that in T140 group.

(3) The author considered that the composition of environmental microbial communities varied greatly under different feeding substrate which effected the bioconversion. However, the different moisture content might be an important factor for the activity of microbes.

Response: In view of the phenomenon in Figure 1, Figure 2 and Figure 3 that "the body length, body width and body weight of BSFL in CK group were higher than those in other experimental groups on the 3 day of bioconversion", we thought it was caused by "the added KDW caused the change of intestinal microbial community of BSFL"

(4) “the feeding substrate of groups of KDW addition (KWTR + sorghum straw + KDW) were different from that the feeding substrate of 1-day old BSFL,” on line 230-231. I could not understand the meaning.

Response: "The feed substrate of 1-day BSFL" was a mistake in translation. "The feed substrate of 1-day BSFL" had been changed to "nursery material". "nursery material" was moist wheat bran for breeding BSFL 3 days before the start of bioconversion.

(5) The author pointed out that the salinity inhibited the activity of BSFL, and they may need 1-3 days to maintain the ion balance. However, the author did not determine the salinity during the process. And different addition of KDW will change the salinity of the feeding substrate. So the explanation should be re-considered.

Response: Based on your previous comments, the entire results and discussion of the article had been rewritten.

(6) What is the “the amount of KDW” mean in Table 4?

Response: This was an error described in a sentence. It had been modified to "Effect of adding KDW on the body components of the BSFL".

Special thanks to you for your good comments.

Reviewer 2 Report

The manuscript titled as 'Effect of Adding Kitchen Deoil Water on the Bioconversion of 2 Kitchen Waste Treatment Residue by Black Soldier Fly Larvae' focused on the soldier fly larvae (BSFL)  as a biological treatment method to treat kitchen waste residues. the authors concluded that The kitchen deoil water could be used as a  nutrient additive in the bioconversion process of KWTR by insect larvae.

The manuscript is well written and some minor adjustments are required.

In introduction section add examples of insects used in bioconversion process with their proper names.

Line 139: Start the sentence with capital letter 'Larvae instead of larvae'.

Line 137: In methodology, please explain the insect with their scientific name and other relevant information.

Make separate headings or sub-headings for explaining the process of P, K and other contents determination.

Replicates and experimental repeats details is missing in methodology section.

Figure legends should be in more detail. 

Assign asterisk or highlight significant findings in the graphical representation

Authors should re-write conclusion and avoid courting results figures/values.

Highlight significant outcome of the study conducted only.

Author Response

Dear reviewer:

Salutation!

First of all, thank you very much for taking time out of your busy schedule to read and revise our article. Thank you for your valuable comments. You have made a comprehensive correction to the structure, content, research methods and results of our paper. It has played a very important role in improving the quality of our article.

We carefully studied the comments of the reviewers and revised the article according to the comments, as follow:

In introduction section add examples of insects used in bioconversion process with their proper names.

Response: The proper name has been added where the article insects first appeared. For example: black soldier fly larvae (Hermetia illucens L.)

Line 139: Start the sentence with capital letter 'Larvae instead of larvae'.

Response: According to your comment, it had been replaced.

Line 137: In methodology, please explain the insect with their scientific name and other relevant information.

Response: Proper nouns of insects and microorganisms had been added to the place where the name first appears. Like: black solider fly larvae (Hermetia illucens L.), Tenebrio molitor, Lactobacillus buchneri et al.

Make separate headings or sub-headings for explaining the process of P, K and other contents determination.

Response: The formula part and the measurement method part of the index in the article had been set up with separate headings.

Replicates and experimental repeats details is missing in methodology section.

Response: Due to the error of expression, the repeating group in the text was actually a parallel group, we rewrote the sentence. The sentence had been modified to "Each experimental group had two parallel groups. The parallel group and the experimental group were conducted at the same time. The conditions of the parallel group were the same as the experimental group."

Figure legends should be in more detail.

Assign asterisk or highlight significant findings in the graphical representation

Response: We had added significant letters in Figure 1, Figure 2 and Figure 3, and re-analyzed them.

Authors should re-write conclusion and avoid courting results figures/values.

Highlight significant outcome of the study conducted only.

Response: Result and Discussion had been rewritten. The main modifications were as follows: the description of data in Result and Discussion had been deleted, and the analysis of experimental results and phenomena in Result and Discussion had been added.

Special thanks to you for your good comments.

Reviewer 3 Report

This manuscript studied the effect of adding Kitchen Deoil Water on the growth, development, and body composition of Black Soldier Fly Larvae. The manuscript is not well written in terms of results. Some sentences are unclear. The English need a complete revision.

Some of my comments and questions are as below:

Lines 20-22: (Notably, 110 mL was considered the best amount for the addition of KDW due to the content of crude protein (CP), crude ash (CA), salinity, total phosphorus (TP), and dry matter (DM) in body composition of BSFL were increased by 3.54%, 6.85%, 0.98%, 0.07%, and 2.98%, respectively, in comparison with those in control.) this sentence needs correction. This is like two sentences which are conflated. Separate them properly.

In line 24, Abstract: Do you mean KDW ?

Line 39-40: the sentence is not clear. Please rewrite it.

Line 43: what does OM stand for? In introduction you need to write the complete names for all abbreviations again.

Line 43 and the next lines:   They were rich in nitrogen….. It should be written in present form.

Line 78: this sentence also needs to be rewritten.

The key of bioconversion was the selection of insects, the studies about the selection of insects were relatively complete at present, such as Kutty et al. [17] used BSFL to degrade kitchen waste.

Line 94: using et al. here is not correct. Please delete it (bioconversion et al.)

Lactobacillus buchneri and other scientific names must be in italic.

Line 118 : change “to explored” to “to explore”

Line 141: “Using a sieve to screen out BSFL of similar size healthy BSFL of 3-day-old of 5 g was selected.” This sentence needs to be rewritten.

Line 145: what is the meaning of CK group?

Line 157: body width was typed twice.

2.4. Analytical Method: in the formula, you do not have w1 and w3 ? you mentioned them in the explanations.

Line 186: what is CA?

Line 191: what is HC1?

Line 204: tables 1 and 2? Are not related to growth and development of BSFL

Lines 205-209: the sentence is too long and it seems that the information presented is not correct according to the figure 1. Please separate the sentence in 2 or more. And check the accuracy of the information. (for example, you do not have T1 group in figures).

Line 213-214: From Figure 3, the body weight of CK group was larger than that of T1, T50, T80, T110 and T140 groups in the 1st to 5th day of bioconversion. I do not think this sentence is correct according to the SE in the figure. The sentence has also grammatical errors.

Line 221-222: the sentence is not correct according to the SE in figures.

Line 223: what do you mean by retared?

you mentioned T1 group in all the results, however, in figures there is no T1

261: Change worm to larva

Line 276: use heavier instead of larger

Line 283: T110 group had the largest, which was 44.86%. this is not correct, because it is not statistically different from T80 and T140

Line 288: About EE content, T1, T50, and T140 groups were significantly smaller than CK group,… this is not correct according to the table 4.

384-387: The Figure 5 showed that T1 and T140 groups of reduction rate were significantly smaller than CK group, T50, T80, T110 and CK groups were no significant difference, T110 group was the closest to CK group, reaching 63.73%, compared with the 71.94% reduction rate of mixed KW in the study by Singh et al. (this sentence is too long and unclear)

Lines 43-48 are one sentence! The sentence is ambiguous. Please divide it into some short sentences to become more clear.

Line 55. It is better to start the sentence without “According to another study”. Using this phrase seems that this sentence must be connected to previous sentence. However, you started it in a new paragraph.

Lines 64-66 need to be rewrite.

Line 66-67 need to be rewrite.

Lines 72-76: it seems that there is no verb for this sentence.

Materials and Methods are not completely described. The authors must categorize the information. They first should describe about the fields and their geographical situation. Then describe the sampling methods.

Line 86: What do you mean from “on every fifth and seventh day” ?

Why did you use 20 fixed wheat plants? Why didn’t use random sampling?

Why did you just count adult apterus aphids? Why not all the aphids(nymphs and adults?)

What do you mean of “depilation yard”?

In figure 2: what do the stars mean?

Line 281: change 3 to three.

Author Response

Dear reviewer:

Salutation!

First of all, thank you very much for taking time out of your busy schedule to read and revise our article. Thank you for your valuable comments. You have made a comprehensive correction to the structure, content, research methods and results of our paper. It has played a very important role in improving the quality of our article.

We carefully studied the comments of the reviewers and revised the article according to the comments.

It must be pointed out that about the grammar of the article, we have tried our best to modify it. However, because of different language usage habits and insufficient English level, there may still be some problems in the revised sentence. we will use the article polishing service provided by the journal to further adjust the article.

Lines 20-22: (Notably, 110 mL was considered the best amount for the addition of KDW due to the content of crude protein (CP), crude ash (CA), salinity, total phosphorus (TP), and dry matter (DM) in body composition of BSFL were increased by 3.54%, 6.85%, 0.98%, 0.07%, and 2.98%, respectively, in comparison with those in control.) this sentence needs correction. This is like two sentences which are conflated. Separate them properly.

Response: This sentence had been modified to “Notably, T110 was considered as the best amount for the addition of KDW. Because compared with the control group, the contents of crude protein (CP), crude ash (CA), salinity, total phosphorus (TP) and dry matter (DM) in body composition of BSFL were increased by 3.54%, 6.85%, 0.98%, 0.07% and 2.98% respectively.”

In line 24, Abstract: Do you mean KDW ?

Response: “KDW concentration” had been modified to “The TK increased with the increase of KDW”

Line 39-40: the sentence is not clear. Please rewrite it.

Response: The part of introduction had been modified. The grammar had been adjusted and the background information with weak relevance in introduction had been deleted.

Line 43: what does OM stand for? In introduction you need to write the complete names for all abbreviations again.

Response: The complete name of OM was organic matter. Since the complete name of the acronym appeared in the abstract, we did not explain the meaning of the acronym in the text. Now we had added the complete name where the acronym first appeared in the text.

Line 43 and the next lines:   They were rich in nitrogen….. It should be written in present form. Response: this error of sentence had been modified.

Line 78: this sentence also needs to be rewritten.

The key of bioconversion was the selection of insects, the studies about the selection of insects were relatively complete at present, such as Kutty et al. [17] used BSFL to degrade kitchen waste.

Response: this sentence had been modified “Selection of insect was one of the key factors in bioconversion. At present, a variety of insects had been used in the recycling of waste.”.

Line 94: using et al. here is not correct. Please delete it (bioconversion et al.)

Response: “et al.” had been deleted.

Lactobacillus buchneri and other scientific names must be in italic.

Response: This error had been modified.

Line 118 : change “to explored” to “to explore”

Response: This error had been modified.

Line 141: “Using a sieve to screen out BSFL of similar size healthy BSFL of 3-day-old of 5 g was selected.” This sentence needs to be rewritten.

Response: This sentence had been modified to “Use a sieve to screen out 5 g of 3-day-old healthy BSFL with similar size”.

Line 145: what is the meaning of CK group?

Response: CK group meant the blank comparison group. The full name had been added where the CK group first appeared.

Line 157: body width was typed twice.

Response: This error had been modified.

2.4. Analytical Method: in the formula, you do not have w1 and w3 ? you mentioned them in the explanations.

Response: W1 and W3 had been deleted, and other formula subscripts had been modified in the order of occurrence.

Line 186: what is CA?

Response: CA was the abbreviation of crude ash. We had added its full name where the text first appeared.

Line 191: what is HC1?

Response: This was an unclear statement. "HCl" had been modified to "mixture of hydraulic acid (HCl) and water (HCl: water=1:3)".

Line 204: tables 1 and 2? Are not related to growth and development of BSFL

Response: Due to the negligence of the writing process, the "figure 1, figure 2 and figure 3" were incorrectly written into "table 1, table 2 and table 3". Now, the "Table 1, Table 2 and Table 3" had been modified to "Figure 1, Figure 2 and Figure 3".

Lines 205-209: the sentence is too long and it seems that the information presented

Response: "T0" in the figure 1, figure 2 and figure 3 were the data of "T1" in the text, but this error was not found during the process of writing the article. Now, "T0" in the figure were modified to "T1".

Line 213-214: From Figure 3, the body weight of CK group was larger than that of T1, T50, T80, T110 and T140 groups in the 1st to 5th day of bioconversion. I do not think this sentence is correct according to the SE in the figure. The sentence has also grammatical errors.

Response: Significant letters had been added to Figure 1, Figure 2 and Figure 3. Figure 1, Figure 2 and Figure 3 had been analyzed and discussed again.

Line 221-222: the sentence is not correct according to the SE in figures.

Response: Significant letters had been added to Figure 1, Figure 2 and Figure 3. Figure 1, Figure 2 and Figure 3 had been analyzed and discussed again.

Line 223: what do you mean by retared?

Response: “retard” means “hinder”.

you mentioned T1 group in all the results, however, in figures there is no T1

Response: T0 in the figure 1, figure 2 and figure 3 were the data of T1 in the text, but this error was not found during the process of writing the article. Now, T0 in the figure were changed to T1.

261: Change worm to larva

Response: This error had been modified.

Line 276: use heavier instead of larger

Response: This error had been modified.

Line 283: T110 group had the largest, which was 44.86%. this is not correct, because it is not statistically different from T80 and T140

Response: We only focus on the largest number and ignore the significance of the difference. According to the significance letter, this sentence had been modified to “but there was no significant difference among T80, T110 and T140 groups”.

Line 288: About EE content, T1, T50, and T140 groups were significantly smaller than CK group,… this is not correct according to the table 4.

Response: Due to the negligence in the writing process, the symbol was used incorrectly. Now "T1" had changed to "T80"."T80" had changed to "T1".

384-387: The Figure 5 showed that T1 and T140 groups of reduction rate were significantly smaller than CK group, T50, T80, T110 and CK groups were no significant difference, T110 group was the closest to CK group, reaching 63.73%, compared with the 71.94% reduction rate of mixed KW in the study by Singh et al. (this sentence is too long and unclear

Response: This sentence had been modified to “The Figure 5 showed that the reduction rate of T1 and T140 groups was significantly smaller than CK group. T50, T80, T110 and CK groups were no significant difference. T110 group reached 63.73%, this reduction rate was lower than the 71.94% reduction rate of mixed KW in Singh et al.”.

Special thanks to you for your good comments.

Round 2

Reviewer 1 Report

Line 89-96 “Kutty et al. [15] used black soldier fly larvae…degradation effect of mixed materials” should be replaced by the name of the insects used in the compost. The concrete context is not necessary.

Three paragraphs in line 74-141 should be refined again to point the most important opinion of the authors. The introduction is not to list the references.

(1) Fig 1-3 is the replication of Table 3. So they should be deleted.

(2) I am still confused about the similar moisture content of CK and T140. In T140 group contained 800ml KDW with additional 980ml KDW. How could it all volatilize in 14 days when the author added the additional KDW every other day! Furthermore, moisture is important during the composting process. It is an important factor to regulate the microbes in the bioconversion process which may change the nutrient content of residue. And obviously, it will have an effect on the degradation of kitchen waste by BSFL. So the moisture should be considered as an important factor, not only “caused by ‘the added KDW caused the change of intestinal microbial community of BSFL’”.

(3) in line 317, the authors said “At this time, the liquid composition of the nursery material was mainly tap water,” so the growth and development of BSFL was the lowest. However, why you did not keep the experiment consistently? Why only the liquid composition of the nursery material was mainly tap water?

(4) “Because some studies showed that [36,37],” why references were positioned here?

(5) small modifications were also shown in the manuscript.

Author Response

Dear Reviewer 1:

Salute!

First of all, thank you very much for taking time out of your busy schedule to read and revise our article again. Your feedback, as well as amendments and replies to the comments are as follows.

(1) Line 89-96 “Kutty et al. [15] used black soldier fly larvae…degradation effect of mixed materials” should be replaced by the name of the insects used in the compost. The concrete context is not necessary.

Response: This sentence has been amended to read:“For example, Kutty et al. [15] used Hermetia illucens L. to degrade KW.”

(2)Three paragraphs in line 74-141 should be refined again to point the most important opinion of the authors. The introduction is not to list the references.

Response: The three paragraphs in the introduction have been refined.

(1) Fig 1-3 is the replication of Table 3. So they should be deleted.

Response: Data that appeared as duplicates in table 3 were removed (e.g., in ending of bioconversion body length, body width, and weight of BSFL.)

(2) I am still confused about the similar moisture content of CK and T140. In T140 group contained 800ml KDW with additional 980ml KDW. How could it all volatilize in 14 days when the author added the additional KDW every other day! Furthermore, moisture is important during the composting process. It is an important factor to regulate the microbes in the bioconversion process which may change the nutrient content of residue. And obviously, it will have an effect on the degradation of kitchen waste by BSFL. So the moisture should be considered as an important factor, not only “caused by ‘the added KDW caused the change of intestinal microbial community of BSFL’”.

Response: Water was one of the important factors that affect the bioconversion of kitchen waste by BSFL, and it was also the direction we were going to study in the future. However, the data of the water content of the feeding substrate of each experimental group after the bioconversion had not been listed in this article, so I don't understand the problem that the water content of the CK group and the T140 group you mentioned was very different. If there was a deviation in my understanding, please tell me the specific stage of your doubts.

(3) in line 317, the authors said “At this time, the liquid composition of the nursery material was mainly tap water,” so the growth and development of BSFL was the lowest. However, why you did not keep the experiment consistently? Why only the liquid composition of the nursery material was mainly tap water?

Response: In the traditional cultivation of BSFL, freshly incubated BSFL were required to be placed into the moist wheat bran for 3 days (Described in 2.2 Experimental Design). The moist wheat bran was dispensed utilizing tap water and dry wheat bran. Deployment of dry wheat bran utilizing tap water is common in the BSFL cultivation process. In addition, about “so the growth and development of BSFL was the lowest. However, why you did not keep the experiment consistently?” so sorry, I don't understand the specific meaning of your question, can you tell me more about your question?

(4) “Because some studies showed that [36,37],” why references were positioned here?

Response: The reference marks have been adjusted to the end of the sentence: ”Some studies showed that when the feeding environment changed, the intestinal microbial community of BSFL also changed; the change in the intestinal microbial community of BSFL was to help BSFL adapt to the new feeding environment [36,37].”

(5) Small modifications were also shown in the manuscript.

Response: Some of the changes came from errors we found. Part of the revision comes from the review opinions of other reviewers.

Reviewer 3 Report

The manuscript is still needed grammar corrections. It was preferred that the authors correct the grammatical errors of the manuscript before sending the revised version. I recommend that the authors use the complete names of CP, CA, TP, DM, etc., in all parts of the manuscript. It is difficult for readers to remember all of these abbreviations. You can use one or two abbreviations.

Some of my comments and questions are as below:

Lines 20-22: (Notably, 110 mL was considered the best amount for the addition of KDW due to the content of crude protein (CP), crude ash (CA), salinity, total phosphorus (TP), and dry matter (DM) in body composition of BSFL were increased by 3.54%, 6.85%, 0.98%, 0.07%, and 2.98%, respectively, in comparison with those in control.) this sentence has not changed in the manuscript.

Line 158-159: “Using a sieve to screen out BSFL of similar size healthy BSFL of 3-day-old of 5 g was selected.” This sentence still needs English correction.

Line 162: (The blank comparison group (CK group). Do you mean control?

Line 173: ” Each experimental group had two parallel groups. The parallel group and the experimental group were conducted at the same time. The conditions of the parallel group were the same as the experimental group. What do you mean by parallel group? Do you mean replications? If yes, you need three replications to analyze your data statistically.

Line 307: what do you mean by “BSFL of T1 group”?

What is T0 in your figure?

In all figures, you have two charts. What is the difference between them? There is no explanation in the figure caption about them. I think you need to delete the above one.

Line 342: what do you mean by “In output (wet basis) and output (wet basis), …..

Author Response

Dear reviewer:

Salutation!

First of all, thank you very much for taking time out of your busy schedule to read and revise our article again. The comments on your feedback and the modification and response to the comments are listed below.

Finally, thank you again for your valuable comments.

The manuscript is still needed grammar corrections. It was preferred that the authors correct the grammatical errors of the manuscript before sending the revised version. I recommend that the authors use the complete names of CP, CA, TP, DM, etc., in all parts of the manuscript. It is difficult for readers to remember all of these abbreviations. You can use one or two abbreviations.

Response: all of abbreviations have been replaced with full names.

Lines 20-22: (Notably, 110 mL was considered the best amount for the addition of KDW due to the content of crude protein (CP), crude ash (CA), salinity, total phosphorus (TP), and dry matter (DM) in body composition of BSFL were increased by 3.54%, 6.85%, 0.98%, 0.07%, and 2.98%, respectively, in comparison with those in control.) this sentence has not changed in the manuscript.

Response: This sentence had been modified to “Notably, the amount of KDW added in T110 group was the most suitable for the growth of BSFL and the accumulation of body composition. Because, compared with the blank comparison group, the content of crude protein (CP), crude ash (CA), salinity, total phosphorus (TP) and dry matter (DM) of BSFL in T110 group increased by 3.54%, 6.85%, 0.98%, 0.07% and 2.98% respectively.”

Line 158-159: “Using a sieve to screen out BSFL of similar size healthy BSFL of 3-day-old of 5 g was selected.” This sentence still needs English correction.

Response: This sentence had been modified to “5g 3-day old BSFL with the same size was screened by sieve.”

Line 162: (The blank comparison group (CK group). Do you mean control?

Response: yes, “The blank comparison group (CK group)” and “control group” meant the same thing. "The blank comparison group (CK group)" had been uniformly used in the latest article.

Line 173: ” Each experimental group had two parallel groups. The parallel group and the experimental group were conducted at the same time. The conditions of the parallel group were the same as the experimental group. What do you mean by parallel group? Do you mean replications? If yes, you need three replications to analyze your data statistically.

Response: Yes, all data in the article (came from figures and tables) was obtained by averaging data from triplicate experiments. This sentence had been modified to “Experiments were replicated three times, each replicate used identical experimental procedures and conditions.”

Line 307: what do you mean by “BSFL of T1 group”?

Response: The expression of this sentence was not clear. It had been modified to “the BSFL of T1 group was lower than that of most experimental groups in body length, body width and body weight.”

What is T0 in your figure?

Response: “T0” in Figure 1, Figure 2 and Figure 3 was “T1” in the article. This error had been corrected.

In all figures, you have two charts. What is the difference between them? There is no explanation in the figure caption about them. I think you need to delete the above one.

Response: Figure 1 showed the two pictures before and after modification in the article, which was mainly to retain the modification trace. Figure 2 and Figure 3 also do the same. In the latest article, the unmodified pictures in Figure 1, Figure 2 and Figure 3 have been deleted, and the correct pictures after modification have been retained.

Line 342: what do you mean by “In output (wet basis) and output (wet basis), ….

Response: This was a mistake in writing, which had now been modified to “In output of BSFL (wet basis) and output of BSFL (dry basis)”